# The Combined Use of Cytokine Serum Values with Laboratory Parameters Improves Mortality Prediction of COVID-19 Patients: The Interleukin-15-to-Albumin Ratio

**DOI:** 10.3390/microorganisms9102159

**Published:** 2021-10-16

**Authors:** Salma A. Rizo-Téllez, Lucia A. Méndez-García, Ana C. Rivera-Rugeles, Marcela Miranda-García, Aarón N. Manjarrez-Reyna, Rebeca Viurcos-Sanabria, Helena Solleiro-Villavicencio, Enrique Becerril-Villanueva, José D. Carrillo-Ruíz, Julian M. Cota-Arce, Angélica Álvarez-Lee, Marco A. De León-Nava, Galileo Escobedo

**Affiliations:** 1Laboratory of Immunometabolism, Research Division, General Hospital of Mexico “Dr. Eduardo Liceaga”, Mexico City 06720, Mexico; sart.17.04@gmail.com (S.A.R.-T.); angelica.mendez.86@hotmail.com (L.A.M.-G.); mm.mirandha7416@gmail.com (M.M.-G.); aaron.manjarrez@gmail.com (A.N.M.-R.); viurcos.reb@hotmail.com (R.V.-S.); 2PECEM, Facultad de Medicina, Universidad Nacional Autónoma de México, Coyoacán, Mexico City 04510, Mexico; 3Laboratory of Oncoimmunology, Biomedical Research Unit, Universidad Nacional Autónoma de México, Tlalnepantla 54090, Mexico; acriverar@unal.edu.co; 4Posgrado de Ciencias Genómicas, Universidad Autónoma de la Ciudad de México, Mexico City 03100, Mexico; helena.solleiro@uacm.edu.mx; 5Laboratory of Psychoimmunology, National Institute of Psychiatry “Ramón de la Fuente”, Mexico City 14370, Mexico; lusenbeve@yahoo.com; 6Research Directorate, General Hospital of Mexico “Dr. Eduardo Liceaga”, Mexico City 06726, Mexico; josecarrilloruiz@yahoo.com; 7Department of Neurology and Neurosurgery, General Hospital of Mexico “Dr. Eduardo Liceaga”, Mexico City 06726, Mexico; 8Facultad de Ciencias de la Salud, Universidad Anáhuac, Campus Norte, Huixquilucan 52786, Mexico; 9Department of Biomedical Innovation, Center for Scientific Research and Higher Education of Ensenada (CICESE), Ensenada 22860, Mexico; jmcotaarce@gmail.com (J.M.C.-A.); ibeth@cicese.mx (A.Á.-L.)

**Keywords:** IL-15, albumin, COVID-19, SARS-CoV-2, mortality, prognosis

## Abstract

Laboratory parameters display limited accuracy in predicting mortality in coronavirus disease 2019 (COVID-19) patients, as with serum albumin. Emerging evidence suggests that cytokine serum values may enhance the predictive capacity of albumin, especially interleukin (IL)-15. We thus investigated whether the use of the IL-15-to-albumin ratio enables improving mortality prediction at hospital admission in a large group of COVID-19 patients. In this prospective cross-sectional study, we enrolled and followed up three hundred and seventy-eight patients with a COVID-19 diagnosis until hospital discharge or death. Two hundred and fifty-five patients survived, whereas one hundred and twenty-three died. Student’s *T*-test revealed that non-survivors had a significant two-fold increase in the IL-15-to-albumin ratio compared to survivors (167.3 ± 63.8 versus 74.2 ± 28.5), a difference that was more evident than that found for IL-15 or albumin separately. Likewise, mortality prediction considerably improved when using the IL-15-to-albumin ratio with a cut-off point > 105.4, exhibiting an area under the receiver operating characteristic curve of 0.841 (95% Confidence Interval, 0.725–0.922, *p* < 0.001). As we outlined here, this is the first study showing that combining IL-15 serum values with albumin improves mortality prediction in COVID-19 patients.

## 1. Introduction

Severe acute respiratory syndrome coronavirus-2 (SARS-CoV-2) is the causal agent of coronavirus disease 2019 (COVID-19), a global pandemic that has affected more than two hundred and thirty-five countries with nearly four million deaths up to June 2021 [1]. Clinical manifestations of COVID-19 are highly heterogeneous, ranging from asymptomatic or mild disease to severe or critical illness in patients who develop acute respiratory distress syndrome (ARDS), sepsis, and multiple organ failure [2]. Case fatality rates of COVID-19 vary depending on the geographical area; however, the vast majority of countries report mortality rates of around 2–4% [3]. Conversely, the case fatality rate of COVID-19 in Mexico is as high as 9.2%, revealing the deep need for novel markers to identify patients at higher risk of death in a timely manner [4].

The combined use of laboratory parameters with inflammatory markers increases the ability to identify COVID-19 patients at a higher mortality risk, as with serum albumin, C-reactive protein (CRP), and neutrophilia. In this sense, the CRP-to-albumin ratio better predicts the severity of COVID-19 than CRP or serum albumin separately [5]. Likewise, the combination of the neutrophil count with serum albumin values improves the area under the receiver operating characteristic (ROC) curve for predicting mortality in COVID-19 patients compared to those found when the neutrophil count or serum albumin are used separately [6]. However, emerging evidence suggests that combining laboratory parameters with markers significantly involved in the cytokine storm may also help to predict the mortality risk in COVID-19 patients, mainly albumin and interleukin (IL)-15 [7,8,9].

Albumin is a plasma protein produced in the liver that exerts multiple physiological functions in blood transport and anticoagulation. Serum albumin is also related to the severity of COVID-19; in fact, patients with the most severe forms of SARS-CoV-2 infection display lower albumin values than patients with the mild-to-moderate disease [10]. Serum albumin is also associated with increased mortality in COVID-19 patients [11]. However, most studies concur that the accuracy of albumin as a mortality predictor in SARS-CoV-2 infection is still limited.

IL-15 is a pleiotropic cytokine expressed by numerous immune and non-immune cells, including monocytes, macrophages, dendritic cells, neurons, epithelial cells, and fibroblasts [12]. IL-15 has a significant role in initiating inflammatory responses against microbial pathogens by modulating innate and adaptive immune cells [13]. A recent study showed that IL-15 serum levels increase in the same proportion as COVID-19 mortality [14]. The use of neutralizing antibodies anti-IL-15 as a potential immunotherapy for patients with severe SARS-CoV-2 infection was recently proposed [15]. Nevertheless, the combined use of IL-15 values with albumin to predict mortality in COVID-19 is unexplored, even though hypoalbuminemia is a common laboratory finding in patients with severe illness, and IL-15 belongs to the cytokine storm that is frequently associated with disease lethality. Thus, the purpose of this study was to examine whether the use of the IL-15-to-albumin ratio allows predicting mortality at hospital admission in a large group of patients with severe SARS-CoV-2 infection.

## 2. Materials and Methods

### 2.1. Patients

Three hundred and seventy-eight patients admitted to the Emergency Department of the General Hospital of Mexico from 30 November 2020 to 9 July 2021, were enrolled in this prospective cross-sectional study. Patients of both sexes were enrolled in the study if they met the following inclusion criteria: 18 years and older, COVID-19 diagnosis confirmed by detection of SARS-CoV-2 specific ribonucleic acid (RNA) in nasopharyngeal swabs using quantitative polymerase chain reaction (qPCR), respiratory distress (>30 breaths per minute), hypoxia (peripheral oxygen saturation <92% on room air), or ≥50% lung involvement on imaging. Patients were excluded from the study if they had a previous diagnosis of human immunodeficiency virus (HIV), hepatitis C virus (HCV), hepatitis B virus (HBV), cancer, endocrine disorders, or autoimmune disease. Pregnant or lactating women and patients under long-term immunomodulatory medication, including non-steroidal anti-inflammatory drugs, were also excluded from the study. All study participants provided written informed consent previously approved by the institutional ethical committee of the General Hospital of Mexico (registration number of the ethical code approval: DI/20/501/03/17). The study rigorously met the principles described in the 1964 Declaration of Helsinki and its posterior amendment in 2013. This cross-sectional study met the Strengthening the Reporting of Observational Studies in Epidemiology (STROBE) Statement: guidelines for reporting observational studies.

### 2.2. Data Collection

We collected demographic and clinical data from the Emergency Department of the General Hospital of Mexico at admission. We also recorded clinical evolution, drug regimen, and inpatient days up to hospital discharge or death. Demographic and clinical data included sex, age, and previous diagnosis of obesity (body mass index (BMI) > 30 kg/m^2^), type 2 diabetes (T2D), hypertension, coronary heart disease (CHD), chronic kidney disease (CKD), and chronic liver disease (CLD).

### 2.3. Laboratory Parameters

We collected laboratory data at admission using the digital version of the electronic health record of the General Hospital of Mexico. Laboratory parameters included albumin, blood glucose, lipid profile, liver function tests, kidney function tests, coagulation markers, hematic biometry, CRP, troponin I, ferritin, procalcitonin, myoglobin, and D-dimer. We measured all laboratory parameters within sixty minutes of the patient’s arrival in the hospital using the Beckman Coulter DxC 700 AU Chemistry Analyzer (Beckman Coulter Inc., Brea, CA, USA), the Coulter LH 780 Hematology Analyzer (Beckman Coulter Inc., Brea, CA, USA), and the BCS^®^ XP System (Siemens Healthcare GmbH, Erlangen, Germany), following standard operating procedures.

### 2.4. IL-15 Serum Levels

At hospital admission, 4 mL blood samples were drawn from all participants and collected in pyrogen-free tubes (Vacutainer^TM^, BD Diagnostics, NJ, USA) at room temperature. After a centrifugation step at 1000 g/4 °C for 30 min, we obtained serum samples for measuring IL-15 in triplicate by the Enzyme-Linked ImmunoSorbent Assay (ELISA) (PeproTech, Cranbury, NJ, USA). We measured IL-15 serum levels within 180 min of the patient’s arrival in the hospital.

### 2.5. Statistics

We collected all demographic, clinical, laboratory, and immune parameters at hospital admission. We followed up with patients until hospital discharge or death. Then, we formed two groups of patients according to the primary outcome: survival or non-survival. In this way, we analyzed and compared all demographic, clinical, and laboratory parameters retrospectively. We used the Shapiro–Wilk test to estimate the normality of data distribution for numerical variables. We compared numerical variables between survival and non-survival groups using unpaired Student’s *T*-tests, showing data as mean ± standard deviation. We used the chi-squared test to analyze categorical variables, showing data as absolute values and percentages. We analyzed ROC curves by obtaining the area under the ROC curve (AUROC) and 95% confidence interval (95% CI), sensitivity, specificity, and odds ratio (OR) for IL-15, serum albumin, and the IL-15-to-albumin ratio as potential mortality predictors. We used the Youden index to calculate optimal cut-off points for IL-15, serum albumin, and the IL-15-to-albumin ratio as potential mortality predictors. The IL-15-to-albumin ratio resulted from dividing the serum levels of IL-15 by albumin. We detected and removed outliers using Grubbs’ test. We considered a *p* value < 0.05 as significant. We used the GraphPad Prism 6.01 software (GraphPad Software, La Jolla, CA 92037, USA), the MedCalc Software (New York, NY 10003, USA), and the IBM SPSS Statistics version 25.0 (IBM, Armonk, NY, USA) for statistical analyses.

## 3. Results

Figure 1 shows an overview of the selection process of eligible participants. After applying the inclusion and exclusion criteria, we enrolled three hundred-sixteen patients in the study. After hospital discharge or death, we retrospectively assigned COVID-19 patients to survival (*n* = 255) or non-survival (*n* = 123) groups (Figure 1).

There were no significant differences between survivors and non-survivors concerning the proportion of women and men (*p* = 0.3) and BMI (*p* = 0.35) (Table 1). Patients in the survival group were, on average, seven years younger than those in the non-survival group (51.4 ± 13.2 versus 58.9 ± 13.7 years, *p* < 0.0001, respectively) (Table 1). There were no significant differences between survivors and non-survivors with respect to the prevalence of obesity, T2D, hypertension, CHD, CKD, and CLD (*p* = 0.8, *p* = 0.1, *p* = 0.5, *p* = 0.3, *p* = 0.6, and *p* = 0.3, respectively). Body temperature and mean arterial pressure showed no changes between survivors and non-survivors (*p* = 0.3 and *p* = 0.1, respectively). Conversely, there were significant differences between survivors and non-survivors for heart rate, breathing rate, and peripheral oxygen saturation (*p* = 0.005, *p* = 0.001, and *p* < 0.001, respectively) (Table 1). Sixty-eight percent (*n* = 84) of non-survivors required admission to the intensive care unit (ICU) for invasive mechanical ventilation, whereas only 23.5% (*n* = 60) of survivors needed ICU admission (*p* < 0.001) (Table 1). The number of inpatient days was, on average, greater for survivors than non-survivors (14.7 ± 9.9 versus 7.3 ± 5.2 days, *p* = 0.014, respectively) (Table 1). All patients enrolled in the study received the same drug regimen: azithromycin, ceftriaxone, oseltamivir, enoxaparin sodium, dexamethasone, and acetaminophen.

Hematological parameters had no significant differences between survivors and non-survivors, except for the neutrophil count, which was 1.7-fold higher in the non-survival group (*p* < 0.001) (Table 2). Overall, there were no significant differences between survivors and non-survivors for most laboratory parameters, including CRP, procalcitonin, fibrinogen, ferritin, and D-dimer. As compared to survivors, non-survivors showed substantial increases in the values of urea, uric acid, aspartate aminotransferase (AST), alkaline phosphatase (ALP), lactate dehydrogenase (LDH), creatinine kinase-MB (CK-MB), and brain natriuretic peptide (BNP) (Table 2).

Non-survivors exhibited significantly lower serum values of albumin than those found in survivors (3 ± 0.5 versus 3.6 ± 0.6 g/dL, respectively, *p* = 0.004) (Figure 2A). IL-15 serum levels showed a significant 1.8-fold increase in non-survivors as compared to survivors (488.7 ± 242.8 versus 261.7 ± 137.6 pg/mL, respectively, *p* < 0.001) (Figure 2B). However, it is worth mentioning that differences between survivors and non-survivors were more evident when circulating values of IL-15 were divided by serum albumin. The IL-15-to-albumin ratio was 2.2-fold higher in non-survivors than in survivors (167.3 ± 63.8 versus 74.2 ± 28.5, respectively, *p* < 0.001) (Figure 2C).

The AUROC for albumin was 0.797 (95% CI, 0.678–0.915, *p* < 0.001), with a cut-off point ≤ 3.3 g/dL, a sensitivity of 90.91%, a specificity of 61.54%, and an OR of 11.31 (95% IC, 2.84–45.06) (Figure 3A). The AUROC for IL-15 was 0.797 (95% CI, 0.675–0.889, *p* < 0.001), with a cut-off point ≤ 364.6 pg/mL, a sensitivity of 68.18%, a specificity of 84.60%, and an OR of 9.79 (95% IC, 2.91–32.98) (Figure 3B). Conversely, a combination between the values of IL-15 and albumin significantly improved the ability to predict mortality in COVID-19 patients. In fact, the AUROC for the IL-15-to-albumin ratio was 0.841 (95% CI, 0.725–0.922, *p* < 0.001), with a cut-off point > 105.4, a sensitivity of 72.73%, a specificity of 87.18%, and an OR of 18.13 (95% IC, 4.81–68.37) (Figure 3C).

The use of IL-15 serum values improved the predictive accuracy of albumin and several clinical and laboratory parameters. For instance, oxygen saturation showed an AUROC of 0.795 (95% CI, 0.647–0.902) for mortality prediction (Figure 4A); however, the combined use of IL-15 values with oxygen saturation exhibited an AUROC of 0.823 (95% IC, 0.705–0.909) (Figure 4B). Likewise, the use of IL-15 significantly increased the AUROC values of the neutrophil count, blood urea, AST, and CRP, among others (Figure 4).

## 4. Discussion

The case fatality rate of COVID-19 has continued to increase in the last few months, especially in countries with slow vaccination rates, such as Mexico [1,4]. Thus, there is still a deep sense of urgency to find novel strategies to help improve our ability to identify COVID-19 patients at high mortality risk. Routine laboratory tests can be measured easily, quickly, and at a low cost, making them good candidates to estimate prognosis after hospital admission. However, the accuracy of laboratory parameters to predict mortality in COVID-19 patients remains limited [16,17]. In this sense, we show that combining laboratory markers’ values with serum cytokines is an excellent strategy to improve the early recognition of COVID-19 patients with an increased risk of death, mainly albumin and IL-15.

Combining cytokine serum values with laboratory parameters recently emerged as a promising approach to estimate prognosis in patients with SARS-CoV-2 infection. A recent study conducted on COVID-19 patients from China demonstrated that the use of IL-2R enhances the accuracy of the lymphocyte count to predict the risk of developing severe-to-critical illness [18]. Likewise, the combined ratio between IL-6 and the T CD8 cell count improves mortality prediction in COVID-19 patients, performing better than other clinical prediction tools, such as the CURB-65 score [19]. In line with previous evidence, we now report for the first time that combining IL-15 with albumin in a single prognostic ratio considerably improved our ability to identify COVID-19 patients at a much higher mortality risk. This trend was observed not only for albumin but also for other clinical and laboratory parameters. In terms of accuracy, the use of IL-15 seemed to enhance the predictive accuracy of albumin concerning that reported using other clinical and inflammatory parameters. A study conducted in 144 COVID-19 patients showed that the neutrophil count-to-albumin ratio (NAR) is an independent predictor of mortality with an AUROC of 0.736 [6]. In parallel, a study conducted in a large group of COVID-19 patients reported that the blood urea nitrogen-to-albumin ratio (BAR) could predict mortality with an AUROC of 0.809 [20]. Likewise, the CRP-to-albumin ratio (CAR) can predict mortality in COVID-19 patients with an AUROC of 0.807 [5]. As we have shown here, the IL-15-to-albumin ratio can predict mortality in COVID-19 patients with an AUROC of 0.841, which improves the predictive accuracy of NAR, BAR, and CAR. Nevertheless, it is crucial to consider that although the IL-15-to-albumin ratio appears to have higher predictive accuracy than other scales, the measurement of routine laboratory parameters is more accessible than IL-15 quantification. We will discuss this notion as part of the limitations of this study.

Since the combined use of the IL-15-to-albumin ratio acts as an accurate mortality predictor, we believe that it is of great importance to discuss the possible mechanisms through which these molecules may contribute to the progression and severity of COVID-19. Decreased serum albumin is one of the most common laboratory alterations in COVID-19 patients that require hospitalization [16,17]. Hypoalbuminemia is also a central component of multiple conditions, such as cancer, cirrhosis, trauma, and sepsis [21,22,23,24]. As a matter of fact, low serum albumin is a mortality predictor in critically ill patients with sepsis and septic shock [25]. Thus, we believe it is crucial to understand the possible mechanisms by which some pathophysiological elements of COVID-19 may down-regulate serum albumin levels. First, decreased hepatic albumin production is associated with increased release of proinflammatory mediators belonging to the COVID-19-related cytokine storm, such as IL-6 and tumor necrosis factor-alpha (TNF-alpha) [26]. Interestingly, injection of Chinese hamster ovary cells transfected with the human gene encoding TNF-alpha into nude mice decreases albumin synthesis [27]. In SARS-CoV-2 infection, Huang and coworkers reported an inverse correlation between IL-6 and serum albumin in patients with poorer survival probability [16]. Moreover, human recombinant IL-6 induces a dose- and time-dependent decrease in the mRNA levels of albumin in in vitro cultured HepG2 cells [28]. This information becomes more significant in patients with COVID-19, wherein increased levels of TNF-alpha and IL-6 may directly decrease albumin production and lead to hypoalbuminemia. In the second place, several acute inflammatory diseases show that serum content of albumin can be redistributed to the interstitial space due to increased vascular permeability and capillary leakage, leading to decreased serum albumin values [26,29]. This information is in line with the fact that COVID-19 is characterized by the release of potent vascular permeability mediators such as arachidonic acid metabolites, IL-8, and monocyte chemoattractant protein-1 (MCP-1), all of which might contribute to the transvascular leakage of albumin [30]. As we have outlined here, the cytokine storm and vascular permeability may act in synergy with SARS-CoV-2 to decrease serum albumin levels in COVID-19 patients, information that might partially explain why hypoalbuminemia appears to be an excellent contributor to mortality prediction in this disease.

In parallel, there is little evidence supporting the possible role of IL-15 in the progression and mortality of COVID-19. After binding to the high-affinity IL-2Rβ/IL-15Rα receptor, IL-15 induces the release of IL-8 and MCP-1 in human monocytes [31]. In severe SARS-CoV-2 infection, IL-8 and MCP-1 can recruit neutrophils and monocytes to the bronchoalveolar space and contribute to tissue damage and respiratory insufficiency [32,33]. In human macrophages, IL-15 can autocrinally promote the release of TNF-alpha, which in turn can induce apoptosis of human coronary artery endothelial cells and bovine pulmonary artery endothelial cells [34,35]. TNF-alpha-induced endothelial cell apoptosis is also associated with endothelial injury, vascular permeability, and systemic capillary leak syndrome, all of which contribute to the progression of COVID-19 [36,37,38]. Thus, IL-15 appears to orchestrate a two-hit deleterious action characterized by an exaggerated inflammatory response and increased endothelial cell apoptosis that together may contribute to the severity of COVID-19. Consistent with this idea, the analysis of 66 soluble biomarkers in 175 patients with severe SARS-CoV-2 infection revealed that IL-15 increases in the same proportion as mortality [14]. A recent cross-sectional study showed that COVID-19 patients with increased serum levels of IL-15 at admission experience a longer duration of hospitalization [39]. Altogether, this information reveals a pivotal role of IL-15 in the progression of SARS-CoV-2 infection and supports using this cytokine as a mortality predictor in COVID-19 patients. As we have outlined here, hypoalbuminemia and IL-15 may share a common pathophysiological mechanism mediated by the cytokine storm, which in turn appears to favor vascular permeability and neutrophil infiltration and leads to increased mortality risk. However, we should further explore the exact molecular mechanism through which albumin and IL-15 worsen survival prognosis in COVID-19.

Another phenomenon captured in our study involves the unexpected finding that there was no difference between survivors and non-survivors for gender and BMI. Most studies have documented that the COVID-19 case fatality rate is commonly higher in men than women [40]. However, a few studies have found that once severe COVID-19 occurs, the risk of dying is similar in women and men. We found no significant differences in the case fatality rate between female and male patients in line with this evidence. Men are more likely to be hospitalized than women; however, mortality is similar in both sexes once severe disease occurs. Likewise, although we noticed that BMI and obesity prevalence tended to be higher in non-survivors than survivors, we found no significant effects on the COVID-19 case fatality rate. Concurring with our findings, several studies have reported that BMI or obesity are not necessarily independent predictors of in-hospital mortality in COVID-19 [41,42,43]. We can explain these controversial findings since, at the time of hospitalization, all patients enrolled in this study had developed the most severe form of COVID-19, including respiratory distress and pneumonia. As mentioned above, most of the studies concur that once severe COVID-19 occurs, the potential contribution of variables such as gender and BMI to mortality rates tends to decrease [40,43]. However, we should consider these findings with caution, and further research is needed to understand the roles of gender and obesity in COVID-19 mortality in a population-specific manner.

Limitations of the study include: (1) the exclusion of HIV, HCV, or HBV seropositive individuals and COVID-19 patients with mild-to-moderate disease, which restricts our findings to a specific group of patients, and (2) the measurement of IL-15 serum levels may not be as easy in primary care. To improve these limitations, we are now conducting additional prospective studies to enroll patients with infectious diseases and other comorbidities. We are also developing rapid IL-15 tests to measure the levels of this cytokine quickly and affordably.

## 5. Conclusions

As we have outlined here, as far as we know, this is the first study demonstrating that the combined use of the IL-15-to-albumin ratio improves mortality prediction in COVID-19 patients that meet hospitalization criteria. Overall, the combination of IL-15 serum values with other clinical and laboratory parameters significantly increased our ability to identify patients at higher mortality risk. The mechanisms through which IL-15 and albumin contribute to mortality in COVID-19 remain to be elucidated. Although albumin measurement appears to be easier than IL-15 quantification in primary care, IL-15 serum levels can be detected quickly and affordably in tertiary healthcare centers, where most patients with severe COVID-19 are admitted. For this reason, we suggest the use of the IL-15-to-albumin ratio with cut-off point >105.4 to triage COVID-19 patients with increased mortality risk, which becomes more relevant in countries with slow vaccination rates. We encourage other research groups to study the impact of combining cytokine serum values with laboratory parameters in the early identification of COVID-19 patients at a much higher mortality risk.

## Figures and Tables

**Figure 1 microorganisms-09-02159-f001:**
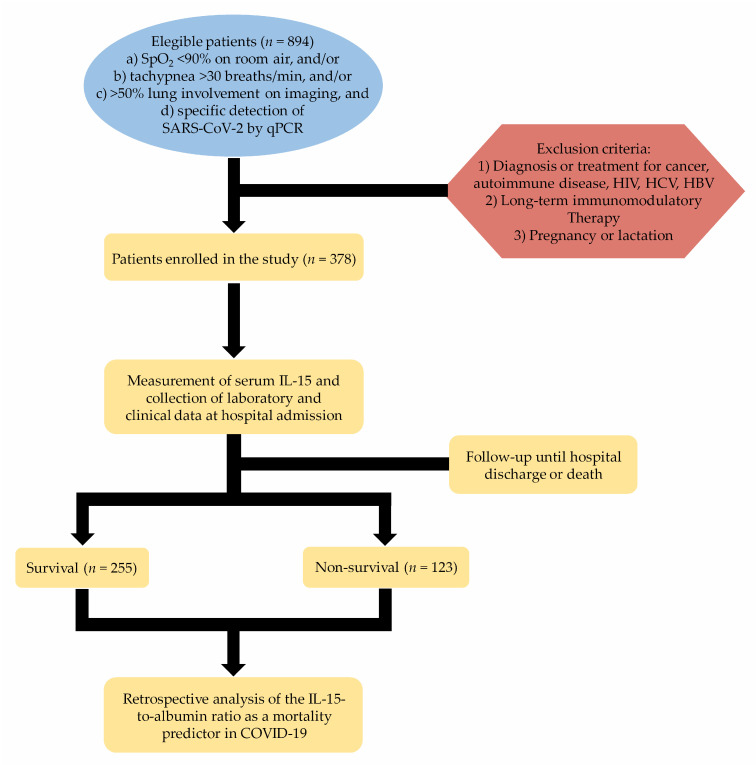
Schematic flow chart illustrating the selection process of eligible patients. SpO2, peripheral oxygen saturation; SARS-CoV-2, severe acute respiratory syndrome coronavirus-2; qPCR, quantitative polymerase chain reaction; HIV, human immunodeficiency virus; HCV, hepatitis C virus; HBV, hepatitis B virus; COVID-19, coronavirus disease 2019.

**Figure 2 microorganisms-09-02159-f002:**
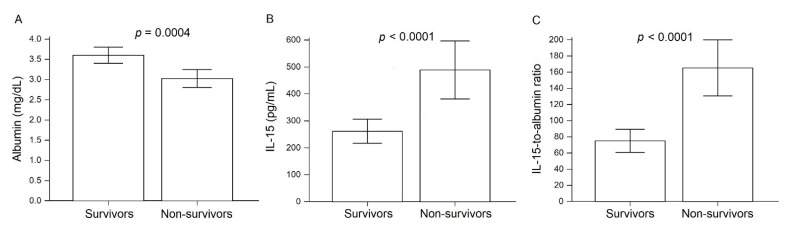
Serum levels of IL-15 and albumin in COVID-19 patients. (**A**) Serum albumin levels in survivors and non-survivors. (**B**) Serum IL-15 levels in survivors and non-survivors. (**C**) The IL-15-to-albumin ratio in survivors and non-survivors. The IL-15-to-albumin ratio resulted from dividing IL-15 serum values by serum albumin. Data are presented as mean ± standard deviation. We considered a *p* value < 0.05 as significant.

**Figure 3 microorganisms-09-02159-f003:**
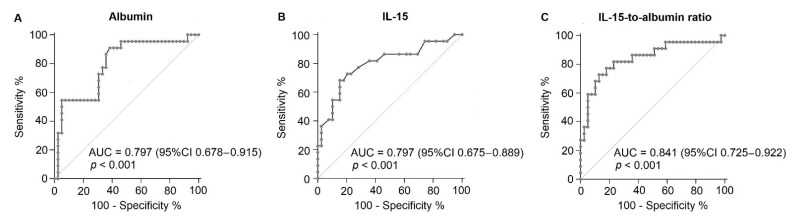
The area under the receiver operating characteristic curves for albumin, IL-15, and the IL-15-to-albumin ratio. (**A**) AUROC for albumin. (**B**) AUROC for IL-15. (**C**) AUROC for the IL-15-to-albumin ratio. We considered a *p* value < 0.05 as significant. AUROC, area under the receiver operating characteristic curves; AUC, area under the curve; CI, confidence interval.

**Figure 4 microorganisms-09-02159-f004:**
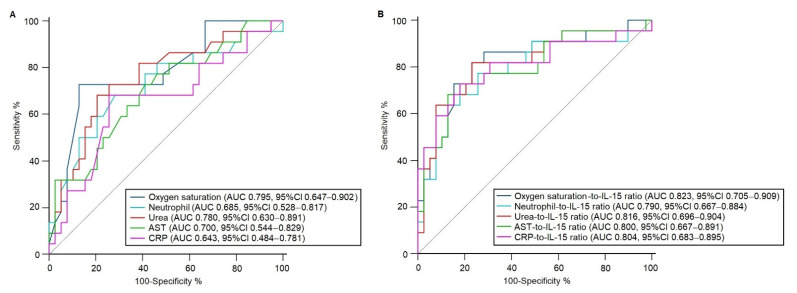
IL-15 serum values improved the area under the receiver operating characteristic curves of clinical and laboratory parameters. (**A**) AUROCs for oxygen saturation, the neutrophil count, blood urea, AST, and CRP. (**B**) AUROCs for the IL-15-to-oxygen saturation ratio, IL-15-to-neutrophil ratio, IL-15-to-urea ratio, IL-15-to-AST ratio, and IL-15-to-CRP ratio. We considered a *p* value < 0.05 as significant. AUROC, area under the receiver operating characteristic curves; AUC, area under the curve; CI, confidence interval; AST, aspartate aminotransferase; CRP, C-reactive protein.

**Table 1 microorganisms-09-02159-t001:** Demographic and clinical characteristics of patients enrolled in the study. Data are expressed as media ± standard deviation or absolute values and percentages. * Differences were considered significant when *p* < 0.05. We only show clinical data that significantly differed between groups. W, women; M, men; BMI, body mass index; T2D, type 2 diabetes; ICU, intensive care unit.

Parameters	Total (*n* = 378)	Survival (*n* = 255)	Non-Survival (*n* = 123)	*p* Value
Gender (W/M)	137/241	97/158	40/83	0.3
Age (years)	54 ± 13.7	51.4 ± 13.2	58.9 ± 13.7	<0.001 *
BMI (kg/m^2^)	26.7 ± 4.8	25.8 ± 6.7	27.5 ± 4.4	0.3
Heart rate (bpm)	89 ± 13.2	87.7 ± 12.9	92 ± 13.5	0.005 *
Breathing rate (bpm)	24.1 ± 4.1	23.6 ± 3.9	25.2 ± 4.5	0.001 *
Oxygen saturation (%)	83 ± 9.5	84.5 ± 8.7	79.8 ± 12.5	<0.001 *
ICU needing (%)	143 (37.8)	60 (23.5)	84 (68.3)	<0.001 *
Inpatient days	12.1 ± 9.2	14.7 ± 9.9	7.3 ± 5.2	0.014 *

**Table 2 microorganisms-09-02159-t002:** Laboratory parameters of patients enrolled in the study. Data are expressed as media ± standard deviation. Differences were considered significant when *p* < 0.05. We only show laboratory parameters that significantly differed between groups. AST, aspartate aminotransferase; ALP, alkaline phosphatase; LDH, lactate dehydrogenase; CK-MB, creatinine kinase myocardial band; BNP, brain natriuretic peptide.

Parameters	Total (*n* = 378)	Survival (*n* = 255)	Non-Survival (*n* = 123)	*p* Value
Neutrophils (×10^3^/mL)	8.6 ± 5.6	6.7 ± 3.9	12 ± 6.7	<0.001
Urea (mg/dL)	59.8 ± 54.7	35.1 ± 14.1	96.8 ± 70.8	<0.001
Uric Acid (mg/dL)	6.2 ± 3.1	5.2 ± 1.7	8 ± 4.3	0.001
AST (IU/L)	37.2 ± 19	33 ± 15.3	46 ± 23	0.008
ALP (IU/L)	93.3 ± 33.2	84.8 ± 28.3	109.9 ± 36.4	0.003
LDH (IU/L)	393.3 ± 224.9	311.2 ± 144.4	539 ± 268.2	0.001
CK-MB (IU/L)	22.1 ± 7.8	19.8 ± 5.6	27 ± 9.9	0.002
BNP (pg/mL)	151.2 ± 304.6	19.1 ± 11.5	291.4 ± 396.4	0.004

## Data Availability

The data presented in this study are available upon request.

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
