# Peer review of "The Combined Use of Cytokine Serum Values with Laboratory Parameters Improves Mortality Prediction of COVID-19 Patients: The Interleukin-15-to-Albumin Ratio"

_microorganisms, 2021, doi:10.3390/microorganisms9102159_

Round 1

Reviewer 1 Report

i red with great interest this report and i found it really intriguing for the daily clinical practice.

yet, i found 2 or 3 point that need an improvement for readers and for take home messages.

  • if IL-15 and albumin ratio have been considered as prognostic marker of overall deaths of inpatients with COVID-19, this should better explained in several section of manuscript (e.g. aim of the study, methods, result and so on)
  • authors should report other illness that are associated to alteration of iL-15 and albumin ration in order to make easy take home message for physicians
  • the study design is clear as reported in the figure but a better definition in the text of best time to detect these values should be performed

Author Response

Reviewer #1

I red with great interest this report and I found it really intriguing for the daily clinical practice. Yet, I found 2 or 3 point that need an improvement for readers and for take home messages.

Reply (R)

We thank to the Reviewer for her/his kind comments on this manuscript.

Query (Q) 1

If IL-15 and albumin ratio has been considered as prognostic marker of overall deaths of inpatients with COVID-19, this should better explain in several section of manuscript (e.g., aim of the study, methods, result and so on).

R1

We respectfully want to clarify that the combined use of the IL-15-to-albumin ratio as a mortality predictor for COVID-19 has been mentioned in the abstract, introduction, materials and methods, results, discussion, and conclusions. Please see this information marked with blue in pages 1-3, 7, 8, and 10.

Q2

Authors should report other illness that are associated to alteration of IL-15 and albumin ration in order to make easy take home message for physicians.

R2

We politely want to clarify that the use of the IL-15-to-albumin ratio has not been reported in other diseases. The IL-15 and albumin, independently, have been widely used in prognosis of several illnesses such as leukemia, sepsis, endocarditis, dermatomyositis, cardiovascular diseases, and colon, gastric and breast cancer, among others (H. Kuniyasu, et.al, Clin Cancer Res. 2003 15;9(13):4802-10; L. Trentin, et.al, Blood. 1996 15;87(8):3327-35; Jekarl DW, et.al, Dis Markers, 2019 8;2019:1089107; Ris T, et.al, Clin Exp Immunol. 2019 196(3):374-382; Takada T, et.al, Respir Med. 2018 Aug;141:7-13; Arques, S, Eur J Intern Med. 2018 52:8-12; Kendall. H, et.al, Biol Res Nurs. 2019 21(3):237-244; Ouyang, X, Clin Lab. 2018 1;64(3):239-245; Fujii, T, et.al, In Vivo. 2020;34(4):2033-2036). However, this is the first study showing the combined use of the IL-15-to-albumin ratio as a mortality predictor, specifically in COVID-19 patients.

Q3

The study design is clear as reported in the figure but a better definition in the text of best time to detect these values should be performed

R3

Serum albumin was detected within sixty minutes of the patient’s arrival in the hospital, while IL-15 was measured within 3 hours of the patient’s arrival in the hospital. Following the Reviewer’s observation, we have added this information in the Materials and Methods section. Please find these changes marked with red in page 3.

We thank you for your very constructive comments on this work. Your criticism has indubitably improved the last version of the manuscript.

Reviewer 2 Report

The authors presented the paper "The combined use of cytokine serum values with laboratory parameters improves mortality prediction of COVID-19 patients: the interleukin-15-to-albumin ratio".

The laboratory parameters in the SARS-2 virus and the COVID-19 are the hot point topic. Any new information is required and can be useful for the medicine.  Below is presented my suggestions and questions for the authors.

1) I think some more information (2 maybe 3 sentences in the introduction) about the albumin have to be present for a better understanding of the functions of the presenting proteins to the journal auditory.

2) One more question, Is it easy to obtain such protein parameters, or maybe the test on one of them will be more suitable/quick enough for the real medicine tests? It can be presented as advantages or limitations of the work in the conclusion section.

3) I think the limitations of the work and the possibility of using such results in the real medicine test have to be presented in conclusions according to so the hard situation of COVID disease. Moreover, the own opinion of the further investigations in this area is required.

4) Some speculations of the possible clear mechanism of these two parameters briefly have to be presented at the end discussion or conclusion section. It may be some author's opinion on smth like that if it is possible.

5) Also, It is an interesting question why the combination of "IL-15 with albumin is an in a single prognostic ratio considerably improved our ability to identify COVID-19 patients at much higher mortality risk". I have seen in the paper interesting results, but not the biochemistry/medicine discussion about it.

"The mechanisms through which IL-15 and albumin contribute to mortality in COVID-19 remain to be elucidated."

Author Response

Reviewer #2

The authors presented the paper "The combined use of cytokine serum values with laboratory parameters improves mortality prediction of COVID-19 patients: the interleukin-15-to-albumin ratio".

The laboratory parameters in the SARS-2 virus and the COVID-19 are the hot point topic. Any new information is required and can be useful for the medicine.  Below is presented my suggestions and questions for the authors.

Reply (R)

We thank to the Reviewer for his/her criticism.

Query (Q) 1

I think some more information (2 maybe 3 sentences in the introduction) about the albumin have to be present for a better understanding of the functions of the presenting proteins to the journal auditory.

R1

Following the Reviewer’s suggestion, we added information regarding albumin functions in the introduction section. Please find this information marked with yellow at page 2.

Q2

One more question, is it easy to obtain such protein parameters, or maybe the test on one of them will be more suitable/quick enough for the real medicine tests? It can be presented as advantages or limitations of the work in the conclusion section.

R2

As the Reviewer correctly pointed out, albumin measurement is easier than IL-15 quantification in clinical practice. However, measurement of IL-15 is not so expensive and can be performed quickly and affordably in tertiary healthcare centers, where most of patients serious-to-critically ill with COVID-19 are admitted. Following the Reviewer’s suggestion, we have added these study limitations at the end of the discussion section. Please find this information marked with pink at page 10.

Q3

I think the limitations of the work and the possibility of using such results in the real medicine test have to be presented in conclusions according to so the hard situation of COVID disease. Moreover, the own opinion of the further investigations in this area is required.

R3

As mentioned, we added a paragraph describing limitations of the study at the end of the discussion section. Also, we added information pointing out the potential use of IL-15 and albumin serum values in clinical practice, having special emphasis on further investigations in this area. Please see this information marked with pink in pages 8 and 9.

Q4

Some speculations of the possible clear mechanism of these two parameters briefly have to be presented at the end discussion or conclusion section. It may be some author's opinion on smth like that if it is possible.

R4

We respectfully want to clarify that the possible mechanisms through which albumin and IL-15 contribute to increased mortality in COVID-19 patients are thoroughly described in the discussion section (please see this information at pages 9 and 10, lines 286-331). However, following the Reviewer’s suggestion, we added a few lines summarizing a speculative mechanism that clarifies the possible contribution of albumin and IL-15 to mortality in SARS-CoV-2 infection. Please find this adding marked with red at page 10.

Q5

Also, It is an interesting question why the combination of "IL-15 with albumin is an in a single prognostic ratio considerably improved our ability to identify COVID-19 patients at much higher mortality risk". I have seen in the paper interesting results, but not the biochemistry/medicine discussion about it.

"The mechanisms through which IL-15 and albumin contribute to mortality in COVID-19 remain to be elucidated."

R5

As stated in the manuscript, the main goal of the study was to assess whether the combined use of IL-15 with albumin in a single prognostic tool improves mortality prediction in COVID-19 patients. However, we concur with the Reviewer that elucidation of the possible mechanisms by which IL-15 and albumin contribute to poor prognosis in SARS-CoV-2 infection is of enormous interest. For this reason, we are now working on elucidating these mechanisms, which will be reported in a different communication.

We sincerely thank to the Reviewer for her/his criticism. We believe that your comments have indubitably improved the last version of the manuscript.

Reviewer 3 Report

The article is very interesting, extensive english editing must be done as well as reorganization of the manuscript and presentation:

-Review abstract formulation

-They included only patients with respiratory distress so no comparisons

- Patients with respiratory distress are those who are hospitalized automatically so there is no difference in surveillance and patients with respiratory distress patients are carefully monitored anyway. So it is disappointing that the authors selected only patients with respiratory distress. They should have compared IL levels with ED patients with and without respiratory distress.

- Explanation why human immunodeficiency virus (HIV), hepatitis C virus (HCV), 90 hepatitis B virus (HBV), cancer, endocrine disorders, and/or autoimmune disease patients were not included

-It is surprising that there is no difference of survival between men and women? This requires an explanation as well as for BMI

-Presentation of table 1 should be different, there is too much information. And this misleads the readership.There should be the title and at the end of the table the legend. There is too much text in the table.

-The line drug regimen is unclear in Table 1 should be reviewed

- As for figure 2, it seems unusual that the results are explained, this should be described as text in the results section and as for figure 3

-It would have been interesting in the discussion section that the authors compare their prognostic score with others described in literature

-The author describes in the discussion why albumin decreases when there is an inflammatory context as well as with interleukine 6. This should have been developed in the introduction when the author explains why he chose these molecules in the first place.

-The discussion should have been oriented on the different prognostic factors, why this score combination is better, and how this could impact clinical practice?

-Moreover, the author should have described the limitations of the manuscript

Author Response

Reviewer #3

The article is very interesting, extensive english editing must be done as well as reorganization of the manuscript and presentation:

Reply (R)

We thank to the Reviewer for her/his very constructive comments on this work. The English version of the manuscript has been edited by Dr. Blair Brown from the University of Minnesota. We have also worked on manuscript reorganization and presentation.

Query (Q) 1

Review abstract formulation

R1

We respectfully want to clarify that abstract formulation is presented according to the journal’s guidelines, wherein background, aims of the study, methods, results, and conclusions can be read without using subheadings.

Q2

They included only patients with respiratory distress so no comparisons

R2

We politely want to clarify that the main goal of the study was to assess the accuracy of the combined use of IL-15 with albumin as a mortality predictor for COVID-19 patients that met hospitalization criteria such as respiratory distress. In fact, this study aims to investigate the use of the IL-15-to-albumin ratio to predict in-hospital mortality in patients with severe COVID-19 but not in ambulatory patients who develop mild-to-moderate symptoms. In parallel, the main comparison we performed was focused on examining retrospectively the IL-15-to-albumin ratio between survivors and non-survivors.

Q3

Patients with respiratory distress are those who are hospitalized automatically so there is no difference in surveillance and patients with respiratory distress patients are carefully monitored anyway. So it is disappointing that the authors selected only patients with respiratory distress. They should have compared IL levels with ED patients with and without respiratory distress.

R3

As mentioned above, the main goal of this study was to examine the accuracy of the IL-15-to-albumin ratio as an in-hospital mortality predictor in patients with severe COVID-19 who developed respiratory distress. We respectfully want to mention that comparison of the IL-15-to-albumin ratio in ED patients that have been hospitalized due to a different cause than severe COVID-19 is not of our interest now.

Q4

Explanation why human immunodeficiency virus (HIV), hepatitis C virus (HCV), 90 hepatitis B virus (HBV), cancer, endocrine disorders, and/or autoimmune disease patients were not included.

R4

We decided not to include patients with active viral infections (HIV, HCV, HBV) and/or non-communicable diseases because it is well known the effect of those on the immune response, especially serum cytokine levels (de Medeiros RM et al., 2016, 23;11(5): e0156163; Spanakis NE et al., 2002, 16(1):40-6; Vayrynen JP et al., 2016, 139:112-121; Al-Humaidi MA et al., 2000, 21(7):639-44, among others). Otherwise, the relationship between COVID-19-related mortality and IL-15 serum levels would have resulted of enormous difficulty to interpret. However, we are now conducting a large multicenter study aimed to assess the accuracy of several mortality predictors in COVID-19 patients with preexisting diseases and data resulting from that study will be reported later.

Q5

It is surprising that there is no difference of survival between men and women? This requires an explanation as well as for BMI.

R5

As the Reviewer correctly pointed out, we were also surprised that there was no difference of survival rates between male and female patients. Despite the vast majority of studies have documented that COVID-19-related mortality is higher in men than women, a few studies have found that the risk of dying is similar in men and women once severe COVID-19 occurs (Raimondi F et al., 2021, 21(1):96). In fact, the COVID-19 case fatality rate is surprisingly higher in women than men in a few countries such as India, Nepal, Vietnam, and Slovenia (Dehingia and Raj, 2021, 9(1):E14-E15). The same applies to BMI, there is increasing number of studies suggesting that BMI is not necessarily predictor of in-hospital mortality in COVID-19 (Nyabera A, et al., 2020, 12(10): e11182; Deng L et al., 2020, 149(e144): 1-10; Chu Y et al., 2020, 25:64, among others). We concur with the Reviewer these apparently controversial findings require an explanation and thus, we have added a paragraph discussing these topics. Please see these changes marked with green at page 10.

Q6

Presentation of table 1 should be different, there is too much information. And this misleads the readership. There should be the title and at the end of the table the legend. There is too much text in the table.

R6

Following the Reviewer’s recommendation, we removed excessive information from presentation of tables 1 and 2. However, please consider that both tables have been prepared according to the journal’s guidelines, where you will find the table legend at the top of table and then the table. Please find these changes marked with yellow at page 5.

Q7

The line drug regimen is unclear in Table 1 should be reviewed

R7

Following the Reviewer’s comment and to avoid misunderstandings, we removed the line drug regimen from Table 1 and added a concise description in the text to clarify that all patients enrolled in the study were subjected to the same drug regimen. Please see these changes marked with pink at page 5.

Q8

As for figure 2, it seems unusual that the results are explained, this should be described as text in the results section and as for figure 3.

R8

Following the Reviewer’s recommendation, we removed excessive information from legends to figures 2, 3, and 4. Please find these changes marked with yellow at pages 7 and 8.

Q9

It would have been interesting in the discussion section that the authors compare their prognostic score with others described in literature

R9

Following the Reviewer’s observation, we added a paragraph comparing the predictive accuracy of the IL-15-to-albumin ratio with other prognostic scores such as NAR, BAR, and CAR. Please see this change marked with pink at pages 8 and 9.

Q10

The author describes in the discussion why albumin decreases when there is an inflammatory context as well as with interleukine 6. This should have been developed in the introduction when the author explains why he chose these molecules in the first place.

R10

Following the Reviewer’s recommendation, we added some context in the introduction section that supports the rationale behind choosing the combination of albumin with IL-15 in a single prognostic ratio to predict mortality of COVID-19 patients. Please see this information marked with yellow and blue at page 2.

Q11

The discussion should have been oriented on the different prognostic factors, why this score combination is better, and how this could impact clinical practice?

R11

As mentioned above, we have now oriented the discussion on the comparison of the IL-15-to-albumin ratio with other prognostic scores, having a special emphasis on highlighting its strengths and weaknesses in the scenario of clinical practice. Please see these changes marked with blue and pink at pages 8 and 9, respectively.

Q12

Moreover, the author should have described the limitations of the manuscript.

R12

Following the Reviewer’s observation, we have added a paragraph describing limitations of the study at the end of the discussion section. Please find this information marked with yellow at page 10.

We sincerely thank to the Reviewer for her/his constructive comments on this work. We believe that your observations and suggestions have indubitably improved the last version of the manuscript.

Round 2

Reviewer 2 Report

Thank you for the revised paper. It is interesting to see the new paper about the mechanism: " For this reason, we are now working on elucidating these mechanisms, which will be reported in a different communication."

One more reason, there is Table 2 with the various parameters. However, I see that for a lot of parameters the error is much more than the average value. But you present a lot of decimal places in the value. For example, 108.73 ± 79.89, better 108.7 ± 79.9, or maybe 108 ± 80

Moreover, It can mean that your sampling value may be wrong outline values. Use carefully the statistics to discarding the values or maybe divide them into several groups. Moreover, you haven't used most of them. Maybe, in the main part of the paper, you can present a small Table with good correlated data and others in SI.

Fig 1 is of poor quality. Maybe, you can improve it.

Author Response

REVIEW REPORT (VERSION 3)

Reviewer #2

Thank you for the revised paper. It is interesting to see the new paper about the mechanism: " For this reason, we are now working on elucidating these mechanisms, which will be reported in a different communication."

Reply (R)

We expect to finish this work by the end of the year. Thanks for your criticism.

Query (Q) 1

One more reason, there is Table 2 with the various parameters. However, I see that for a lot of parameters the error is much more than the average value. But you present a lot of decimal places in the value. For example, 108.73 ± 79.89, better 108.7 ± 79.9, or maybe 108 ± 80

R1

Attending to your recommendation, we annotated all values using one decimal. Furthermore, we substantially reduced the length of tables by deleting parameters that showed no significant differences. Please see these changes in Table 2 (pages 5 and 6).

Q2

Moreover, It can mean that your sampling value may be wrong outline values. Use carefully the statistics to discarding the values or maybe divide them into several groups. Moreover, you haven't used most of them. Maybe, in the main part of the paper, you can present a small Table with good correlated data and others in SI.

R2

Following your recommendation, we looked for outliers by the Grubbs’ test and reanalyzed Table 2 (along with all manuscript data, including albumin and IL-15 serum levels). Moreover, we chose to restrict Table 2 to parameters with significant differences. As you will be able to see, most of the variables now exhibit a standard deviation value less than the average value, except for BNP, where nine patients in the non-survivor group showed BNP serum levels more than ten times the average. Please see these changes in Table 2 and the rest of the manuscript text, especially page 3 marked with yellow. We thank you for your recommendations that have indubitably improved the last version of the manuscript.

Q3

Fig 1 is of poor quality. Maybe, you can improve it.

R3

As you requested, we improved the quality of figure 1. Please see this change in page 4.

Reviewer 3 Report

The modifications in the manuscript improve overall comprehension and bring more clarity.

The tables should be simplified

Author Response

REVIEW REPORT (VERSION 3)

Reviewer #3

The modifications in the manuscript improve overall comprehension and bring more clarity.

Reply (R)

Thanks for your criticism that has indubitably improved the last version of the manuscript.

Query (Q) 1

The tables should be simplified.

R1

Attending to your recommendation, we substantially simplified the content of Tables 1 and 2 by only showing data with significant differences. Now, Tables 1 and 2 are easy to read and compare. Please see these changes on pages 5 and 6.
